# Cross-Recognition of Promoters by the Nine SigB Homologues Present in *Streptomyces coelicolor* A3(2)

**DOI:** 10.3390/ijms22157849

**Published:** 2021-07-22

**Authors:** Beatrica Sevcikova, Bronislava Rezuchova, Vladimira Mazurakova, Dagmar Homerova, Renata Novakova, Lubomira Feckova, Jan Kormanec

**Affiliations:** Institute of Molecular Biology, Slovak Academy of Sciences, 845 51 Bratislava, Slovakia; beatrica.sevcikova@savba.sk (B.S.); bronislava.rezuchova@savba.sk (B.R.); vbenkovsmaz@gmail.com (V.M.); dagmar.homerova@savba.sk (D.H.); renata.novakova@savba.sk (R.N.); lubomira.feckova@savba.sk (L.F.)

**Keywords:** differentiation, promoter, regulation, sigma factor, *Streptomyces*, stress response

## Abstract

In contrast to *Bacillus subtilis*, *Streptomyces coelicolor* A3(2) contains nine homologues of stress response sigma factor SigB with a major role in differentiation and osmotic stress response. The aim of this study was to further characterize these SigB homologues. We previously established a two-plasmid system to identify promoters recognized by sigma factors and used it to identify promoters recognized by the three SigB homologues, SigF, SigG, and SigH from *S. coelicolor* A3(2). Here, we used this system to identify 14 promoters recognized by SigB. The promoters were verified in vivo in *S. coelicolor* A3(2) under osmotic stress conditions in *sigB* and *sigH* operon mutants, indicating some cross-recognition of these promoters by these two SigB homologues. This two-plasmid system was used to examine the recognition of all identified SigB-, SigF-, SigG-, and SigH-dependent promoters with all nine SigB homologues. The results confirmed this cross-recognition. Almost all 24 investigated promoters were recognized by two or more SigB homologues and data suggested some distinguishing groups of promoters recognized by these sigma factors. However, analysis of the promoters did not reveal any specific sequence characteristics for these recognition groups. All promoters showed high similarity in the -35 and -10 regions. Immunoblot analysis revealed the presence of SigB under osmotic stress conditions and SigH during morphological differentiation. Together with the phenotypic analysis of *sigB* and *sigH* operon mutants in *S. coelicolor* A3(2), the results suggest a dominant role for SigB in the osmotic stress response and a dual role for SigH in the osmotic stress response and morphological differentiation. These data suggest a complex regulation of the osmotic stress response in relation to morphological differentiation in *S. coelicolor* A3(2).

## 1. Introduction

Gram-positive bacteria of the genus *Streptomyces* are one of the best producers of biologically active secondary metabolites with a wide range of activities (antibacterial, antitumor, antifungal, herbicidal, immunosuppressive, and anthelmintic). These filamentous bacteria undergo a complex process of morphological differentiation. This process is associated with the so-called physiological differentiation, which represents the production of these secondary metabolites. Morphological differentiation usually takes place in surface-grown strains on solid agar media and is characterized by several developmental stages. Some strains (e.g., *Streptomyces venezuelae*) may also differentiate in liquid-grown cultures. It begins by germinating spores to form a network of branched multinucleoidal hyphae growing by tip extension (so-called vegetative substrate mycelium). In response to various signals, special reproductive white air-grown hyphae (so-called aerial mycelium) emerge from the substrate mycelium. Various genetic, cytological, and biochemical approaches have indicated that a programmed cell death process takes place in the substrate mycelium to provide nutrients for aerial mycelium growth. These aerial hyphae similarly grow by tip extension. After their growth ceases, they undergo a process of synchronous septation to unigenomic pre-spore compartments, which finally maturate into spores with a characteristic color [1,2].

Streptomycetes are exposed to various nutritional and environmental stresses in their natural habitats. The regulation of morphological and physiological differentiation is associated with these stress signals [3,4]. In many bacteria, the response to these stresses is mediated by alternative sigma factors of RNA polymerase, which govern the expression of genes encoding the stress response proteins necessary to overcome these adverse conditions. The best characterized example is the general stress response sigma factor SigB in the Gram-positive bacterium *Bacillus subtilis*. Its gene is located in the *rsbV*, *rsbW*, *sigB*, *rsbX* operon along with genes encoding the regulators of its activity. Under non-stress conditions, low operon expression is provided by a promoter recognized by RNA polymerase with the principal sigma factor SigA. However, under stress conditions, the operon is autocatalytically activated by a promoter recognized by RNA polymerase with SigB. In addition, SigB activity is regulated by the so-called partner-switching phosphorylation mechanism with anti-sigma factor RsbW, anti-anti-sigma factor RsbV, two PP2C phosphatases RsbP and RsbU, and feedback phosphatase RsbX. Under non-stress conditions, SigB is inactive because it is sequestered with RsbW. In addition, RsbW phosphorylates, thereby inactivating RsbV, which is unable to interact with RsbW. Upon stress activation, phosphorylated RsbV is dephosphorylated by two PP2C-type phosphatases from two different stress pathways; RsbP detects nutritional stress and is activated by the binding of the RsbQ protein, and RsbU is activated by various environmental stresses via the RsbT activator through a large “stressosome” complex transducing these stresses. Dephosphorylated RsbV replaces RsbW from its complex with SigB, and the released SigB can interact with core RNA polymerase to form a holoenzyme that directs the expression of the SigB regulon stress response genes. The feedback control after the stress conditions have subsided is provided by RsbX phosphatase. It dephosphorylates the components of the stressosome, leading to deactivation of RsbT activator [5,6,7].

The genomic sequence of the best studied model of the *Streptomyces* strains, *Streptomyces coelicolor* A3(2), revealed genes for up to 65 different sigma factors [8]. Phylogenetic analysis revealed nine close homologues of *B. subtilis*, SigB (SigB, SigF, SigG, SigH, SigI, SigK, SigL, SigM, SigN). They contain the highly conserved regions 2.4 and 4.2, involved in the interactions with the promoter regions -10 and -35, respectively. In addition, regulation of these SigB homologues appears to be more complex than in *B. subtilis*. *S. coelicolor* A3(2) even contains 45 homologues of anti-sigma factor RsbW, 17 homologues of anti-anti-sigma factor RsbV, and 44 homologues of PP2C phosphatases RsbU/RsbP [7,9].

Characterizations of these SigB homologues revealed their dominant role controlling morphological differentiation and the osmotic stress response. The first characterized SigB homologue was SigF (SCO4035). Phenotypic analysis of its *sigF* mutant revealed its key role in the late stages of morphological differentiation, in the process of spore maturation and pigmentation [10]. Its monocistronic *sigF* gene is expressed during sporulation and is spatially located in the prespore compartment of sporulating aerial hyphae [11,12]. The *sigF* gene is not located on the operon with genes encoding its regulator. However, one of the 45 homologues of RsbW, RsfA (SCO4677), was found to interact with SigF, suggesting that it may be an anti-sigma factor for SigF [13]. Only two target genes under the SigF control have been identified in *S. coelicolor* A3(2): the *whiEVIII* gene for spore pigment biosynthesis, and the *sspA* gene for secreted lipoprotein, which plays a role in sporulation [14,15]. Interestingly, immediately downstream of the *sigF* gene there is a gene encoding other SigB homologues, SigN (SCO4034). This sigma factor plays a role in morphological differentiation in the subapical stem region of aerial hyphae. The expression of the *sigN* gene is driven by two tandem promoters upregulated during aerial mycelium formation. Its only identified target gene, *nepA*, encoding a small extracellular protein, is spatially expressed in this new compartment [16]. Another SigB homologue, SigK (SCO6520), plays a negative role in morphological differentiation and secondary metabolism. The expression of its monocitronic gene, *sigK*, gradually increased during differentiation. However, the level of SigK is post-translationally regulated, and decreases during the formation of aerial hyphae [17]. No specific role could be assigned to another SigB homologue, SigG (SCO7341), after phenotypic analysis of its mutant. Its monocitronic gene was not expressed during differentiation nor under several stress conditions [18]. However, its gene was expressed during spore germination, suggesting its role in this process [19]. Two SigG-dependent promoters were identified using the heterologous *E. coli* two-plasmid system [20]. On the other hand, another SigB homologue, SigI (SCO3068), is involved in the osmotic stress response. Transcription of its monocistronic *sigI* gene is driven by the osmotically induced *sigIp* promoter. Two regulators, anti-sigma factor PrsI and anti-anti-sigma factor ArsI, encoded by divergent genes upstream of the *sigI* gene, are involved in SigI regulation by the partner-switching phosphorylation mechanism [21]. Two other characterized SigB homologues, SigB (SCO0600) and SigH (SCO5243), are thought to play a dual role in morphological differentiation and the osmotic stress response. The insertional *sigB* mutant affected aerial mycelium formation, and was sensitive to osmotic stress conditions. Its gene is located in the *rsbB*, *rsbA*, *sigB* operon. The *rsbB* gene encodes a homologue of anti-anti-sigma factor RsbV, and *rsbA* encodes a homologue of anti-sigma factor RsbW. The operon is driven by two promoters: a constitutive *sigBp2* controlling the entire operon, and an osmotically-induced *sigBp1* directing only *sigB*, which was partially dependent upon SigB. However, only RsbA is involved in the regulation of SigB. In addition to RsbA, another homologue of RsbV (SCO7325) is involved in the partner-switching phosphorylation mechanism to activate SigB [22,23]. SigB activity is also regulated by another OsaC regulator (SCO5747), which is required to return the *sigB* expression to basal level after overcoming osmotic stress [24]. DNA microarray analysis under osmotic stress conditions revealed more than 280 osmotic stress induced genes differentially dependent on SigB. They encode proteins participating in osmoregulation and oxidative stress response. In addition, SigB acts as a master osmotic stress regulator and directs the expression of genes for two other SigB homologues, SigL and SigM, under osmotic stress conditions. Phenotypic analysis of their mutants suggests the role of SigL in sporulation of aerial hyphae and SigM in efficient sporulation [25]. SigM has been shown to interact with the anti-sigma factor RsmA (SCO7313), which modulates its activity through the iron-sulfur [2FE-2S] cluster [26]. Another SigB homologue with a proposed dual role in the osmotic stress response and differentiation is SigH. The *sigH* mutant affected both processes, growth under high osmotic conditions and septation of aerial hyphae [27]. Its gene is located in the *ushY*, *ushX*, *sigH* operon, together with the gene encoding its specific anti-sigma factor UshX (SCO5244). The expression of the operon is controlled by four promoters at the different developmental stages and under different stress conditions [28,29]. The operon is autoregulated by SigH under osmotic stress conditions through the *sigHp2* promoter [27]. In addition, SigH is also regulated at the post-translational level. The *sigH* gene is translated to a larger protein, SigH52/51, mainly present in the exponential phase or in the early stages of development, and its shorter form, SigH37, mainly present in the stationary phase and during later stages of development. In addition, the larger form undergoes proteolysis to the N-terminally truncated forms SigH38 and SigH34 during the stationary phase and at the later stages of development [30]. SigH is negatively regulated by its specific anti-sigma factor UshX/PrsH [27,30,31]. Activation of SigH under osmotic stress conditions is ensured by sequestration of UshX with anti-anti-sigma factor BldG (SCO3549) [32]. This regulation is more complex because, in addition to UxhX, BldG also interacts and is phosphorylated by several RsbW homologues. Two of them, RsfA and SCO7328, were found to interact with other SigB homologues (SigF, SigG, SigK, SigM), suggesting a complex and interconnected regulation of the stress response and differentiation in *S. coelicolor* A3(2) [33]. DNA microarray analysis revealed that transcription of six genes encoding sigma factor of the SigB family (*sigB*, *sigH*, *sigI*, *sigK*, *sigL*, *sigM*) was induced by osmotic stress in *S. coelicolor* A3(2) [34].

In this study, we attempted to further characterize these nine SigB homologues in *S. coelicolor* A3(2). Previously, we developed an efficient *E. coli* two-plasmid system for the identification of promoters directly recognized by sigma factors [35]. The system was used to identify promoters recognized by SigF, SigG and SigH from *S. coelicolor* A3(2) [20,36,37,38,39]. In the present study, we used this system to identify promoters recognized by SigB from *S. coelicolor* A3(2). The transcriptional activity of the identified promoters was tested in vivo in *S. coelicolor* A3(2) under osmotic stress conditions in two newly prepared *sigB* and *sigH* operon mutants, indicating some cross-recognition of the promoters by these two SigB homologues. Interestingly, the two identified SigB-dependent promoters were previously identified by similar two-plasmid screening with the sigma factors SigG and SigH. It further suggested this cross-recognition of promoters by several SigB homologues in *S. coelicolor* A3(2). Therefore, we used this two-plasmid system to analyze the recognition of all identified SigB-, SigF-, SigG-, and SigH-dependent promoters with all nine SigB homologous sigma factors from *S. coelicolor* A3(2). The results clearly confirmed this cross-recognition. In addition, we investigated the presence of both sigma factors SigB and SigH under osmotic stress conditions and during differentiation by immunoblot analysis.

## 2. Results and Discussion

### 2.1. Levels of SigB and SigH after Osmotic Stress and during Differentiation

Of the nine SigB homologues in *S. coelicolor* A3(2), only two, SigB and SigH, contain an operon with a validated anti-sigma factor gene, *rsbB*, *rsbA*, *sigB* [22,23] and *ushY*, *ushX*, *sigH* [28,31], partially similar to the *B. subtilis sigB* operon. In addition, they are thought to play a dual role in morphological differentiation and the osmotic stress response. To examine the presence of both sigma factors, SigB and SigH, under osmotic stress conditions and during differentiation, we prepared polyclonal antibodies against both sigma factors. In addition, we also prepared a polyclonal antibody against SigH-specific anti-sigma factor UshX. The presence of all three proteins was investigated in defined liquid and solid minimal medium with mannitol as a carbon source. These minimal media have defined morphological phases [11] and osmolarity, in contrast to the rich media used previously for immunodetection of SigH [30] or SigB [25].

Immunoblot with a SigH-specific antibody using crude extracts from *S. coelicolor* M145 and *S. coelicolor* Δ*sigHop* operon mutant grown in liquid minimal medium NMP and under osmotic stress conditions revealed the presence of the three previously described forms of SigH (52/51 kDa, 38/37 kDa, 34 kDa) [30], in exponentially grown cells. As expected, these proteins were absent from the *S. coelicolor* Δ*sigHop* operon mutant (Figure 1a). The level of shorter 38/37 and 34 kDa forms increased in the stationary phase. The levels of both 52/51 and 38/37 kDa forms were slightly induced under osmotic stress conditions. These results were consistent with our previous transcriptional data, where the *ushY*, *ushX*, *sigH* operon was driven by four differentially expressed promoters. Transcription from the *sigHp1* promoter increased in the stationary phase and transcription from the *sigHp2* promoter was induced by osmotic stress conditions [28].

Interestingly, using identical crude extracts, we did not detect any signal corresponding to UshX (theoretical M_r_ = 14,273) in the immunoblot with the UshX-specific antibody (Figure 1a), whereas these antibodies detected UshX overproduced in *E. coli* (16,454). This suggested specific proteolytic degradation of this anti-sigma factor in *S. coelicolor* M145 liquid cultures.

Immunoblot with a SigB-specific antibody using crude extracts from *S. coelicolor* M145 and *S. coelicolor* Δ*sigBop* operon mutant strains grown under similar conditions revealed a single 39 kDa form of SigB that was absent in the *S. coelicolor* Δ*sigBop* operon mutant (Figure 1a). Its level increased slightly in the stationary phase but was highly induced in conditions of osmotic stress. These results were consistent with previously published immunoblot data [25]. The estimated molecular weights of SigH and SigB were higher than M_r_, calculated from the amino acid sequence. This abnormal mobility on SDS-PAGE is a hallmark of sigma factors. They usually migrate more slowly due to their positive and negative clusters [40].

The presence of both sigma factors was similarly examined during morphological differentiation using crude extracts prepared from surface-grown cultures of *S. coelicolor* on solid minimal MM medium containing 0.5% mannitol to different developmental stages. Cultures were harvested at previously defined time points that correspond to different developmental stages of morphological differentiation: vegetative substrate mycelium, aerial hyphae formation, and the early and late stages of sporulation [11]. At the time of collection, they were examined microscopically. Consistent with previous results [30], the level of the 52/51 kDa kDa form of SigH decreased during differentiation, and the levels of the shorter forms (38/37 kDa and 34 kDa) increased, reaching their highest levels at day 5, corresponding to sporulation of aerial hyphae. All signals were correctly absent in the *S. coelicolor* Δ*sigHop* operon mutant (Figure 1b). In contrast to the results from cultures grown in liquid NMP medium, where UshX was absent, the level of UshX increased continuously during differentiation. The signal corresponding to the theoretical mass of UshX (14,273) was missing in the *S. coelicolor* Δ*sigHop* operon mutant.

Interestingly, SigB could not be detected in identical crude extracts from surface-grown *S. coelicolor* M145 (Figure 1b). However, when crude extracts were prepared from similar surface-grown cultures of *S. coelicolor* M145 and *S. coelicolor* Δ*sigBop* grown in minimal MM medium and MM medium with 0.5 M NaCl, immunoblot with the SigB-specific antibody revealed a weak signal corresponding to SigB only being present in conditions of osmotic stress and not in *S. coelicolor* Δ*sigBop* (Figure 1b).

Based on these results, we can conclude that SigB is present mainly under conditions of osmotic stress and is absent during differentiation. Therefore, it plays a dominant role in the osmotic stress response. However, in contrast to previous results [22], it probably has no role in morphological differentiation. In contrast, SigH probably plays a dual role in both the osmotic stress response and differentiation, as already indicated [27,29]. Its level, together with the translationally coupled anti-sigma factor UshX, increases during differentiation with the highest levels during aerial mycelium septation. However, it is also weakly induced during the osmotic stress conditions during growth in the liquid minimal medium.

Interestingly, SigH-specific anti-sigma factor UshX is not present in liquid-grown cells, even under osmotic stress conditions, although its gene is translationally coupled with *sigH* and was transcribed at these conditions [28]. Therefore, this anti-sigma factor is probably specifically proteolytically degraded in liquid-grown *S. coelicolor* M145 cultures. The reason for this specific degradation is unclear. This may be related to another proposed role of SigH in osmotic stress response during cultivation in liquid medium [27,28]. As both UshX and SigH are produced under these conditions, UshX is degraded to provide SigH activity to recognize osmotically-induced promoters, including the *sigHp2* promoter itself, under osmotic stress conditions [27,28]. Such degradation is not required during morphological differentiation, where the levels of both UshX and short versions of SigH gradually increase towards later stages of development (Figure 1b); but UshX is sequestered by its specific anti-anti-sigma factor BldG, to ensure SigH activation [32].

### 2.2. Deletion of the sigB and sigH Operons in S. coelicolor A3(2)

To study the regulatory role of the master osmotic stress-response sigma factor SigB in *S. coelicolor* A3(2), we deleted the entire *rsbB*, *rsbA*, *sigB* operon by the REDIRECT strategy [41] in the wild-type *S. coelicolor* M145 strain (Appendix A). However, in contrast to previous results, where deletion of *sigB* dramatically affected morphological differentiation lacked aerial mycelium formation, and formed bald colonies [22], the *S. coelicolor* Δ*sigBop* operon mutant was similar to the wild-type *S. coelicolor* M145 strain. In addition, we examined the phenotype under osmotic stress conditions. The *S. coelicolor* Δ*sigBop* operon mutant was only partially affected, in contrast to the previously published *sigB* mutant which did not grow under osmotic stress conditions [22]. Using a similar strategy, we deleted only the *sigB* gene in *S. coelicolor* M145. However, the phenotype of the *sigB* mutant was similar to the *sigBop* operon mutant. Its spores were only slightly paler (Appendix A). Differences between the phenotype of our *sigB* mutant and previously published *sigB* mutant [22] may be due to a different genetic background or disruption strategy. Our results suggest that *sigB* is unlikely to be involved in differentiation. It plays a dominant role in response to the osmotic stress response. It is also consistent with the absence of SigB during differentiation (Figure 1b).

Another strategy was used to delete the entire *ushY*, *ushX*, *sigH* operon in the wild-type *S. coelicolor* M145 strain (Appendix A). The *S. coelicolor* Δ*sigHop* mutant was phenotypically similar to the wild-type *S. coelicolor* M145 strain. In addition, we examined the phenotype under osmotic stress conditions, and the *S. coelicolor* Δ*sigHop* operon mutant was also similar to the wild-type *S. coelicolor* M145 strain. It is consistent with the previously published phenotype of the *sigH* mutant strain in *S. coelicolor* A3(2) [30]. However, this phenotype of the *S. coelicolor* Δ*sigHop* operon mutant differed slightly from our previously prepared deleted *sigH* mutant in *S. coelicolor* M145, which formed pale grey sporulation with less abundant aerial hyphae septation, although spore morphology was not affected [27]. Based on current data, we can conclude that the deletion of the entire *ushY*, *ushX*, *sigH* operon had no effect on morphological differentiation nor osmotic stress response. However, it cannot be rule out that the *sigH* operon affects expression of genes in response to osmotic stress, although this does not visibly affect the phenotype.

### 2.3. Identification of SigB Regulon Using a Heterologous Two-Plasmid System

We have previously developed a heterologous system to identify promoters directly recognized by sigma factors [35]. The method is based on two compatible plasmids in *E. coli*. One plasmid carries the sigma factor gene under the control of the inducible *trc* promoter, and the other plasmid contains a library of *Streptomyces* chromosomal fragments upstream of the *lacZα* reporter gene. Under inducing conditions, the heterologously produced sigma factor can associate with *E. coli* core RNA polymerase, and the resulting RNA polymerase holoenzyme is able to recognize its cognate promoter in the library of DNA fragments and activate the *lacZ*α reporter gene. Such colonies were blue on 5-bromo-4-chloro-3-indolyl-β-D-galactopyranosid (X-gal) selective plates. To distinguish between promoters constitutively active in *E. coli* and sigma factor-dependent promoters, plasmids isolated from such blue colonies were subsequently transformed into two *E. coli* strains; one strain with the expression plasmid alone (pAC5mut2), and the other with the expression plasmid containing the particular sigma factor gene. The sigma factor-dependent promoter is active only in *E. coli* with the expression plasmid containing the sigma factor gene. This strategy is illustrated for the *S. coelicolor sigB* gene in Figure 2. This heterologous two-plasmid system has been successfully used to identify promoters recognized by the sigma factors SigF, SigG and SigH from *S. coelicolor* A3(2) [20,36,37,38,39], SigB from *Staphylococcus aureus* [42], and SigF from *Mycobacterium tuberculosis* [43].

In the present study, we used this method to identify promoters recognized by SigB from *S. coelicolor* A3(2), as shown in Figure 2. Two *S. coelicolor* M145 libraries with small DNA fragments (0.5–1 kb *Taq*I cloned into the *Cla*I site of pSB40N and *Sau*3AI cloned into the *Bam*HI site of pSB40N) were used to transform *E. coli* containing pAC-sigB. Screening of about 150,000 colonies from each library under inducing conditions on LBACIX plates revealed 29 SigB-dependent clones in the *Taq*I library (pBT) among the 710 blue colonies analyzed, and 18 in the *Sau*3AI library (pBS) among 577 blue colonies analyzed. Nucleotide sequence analysis of all plasmids isolated from positive clones revealed 14 different plasmids (Figure 3). Most of them were present in both libraries. To identify the transcription start site (TSS) of the SigB-dependent promoters, high-resolution S1-nuclease mapping was performed using RNA isolated from *E. coli* containing each positive plasmid and pAC-sigB grown under inducing conditions. *E. coli* containing each positive plasmid and the expression vector pAC5mut2 alone was used as a negative control. An RNA-protected fragment was identified for each plasmid present only in *E. coli* with pAC-sigB (Figure 3a), corresponding to the TSS for each promoter (Figure 3b). The promoter sequences were similar to the consensus sequence of homologous sigma factor SigB from *B. subtilis* [44] and the previously proposed consensus sequence of SigB-dependent promoters in *S. coelicolor* A3(2) [45].

The identified SigB-dependent promoters were located upstream of the genes or operons they control (Table 1). The *SCO1089p* promoter controls operon expression (*SCO1089*, *SCO1088*, *SCO1087*, *SCO1086*) and all four genes were previously identified as SigB-dependent under osmotic stress conditions using a DNA microarray in *S. coelicolor* J1501 [25]. In addition, three monocistronic genes driven by the identified promoters (*SCO2315*, *SCO5998*, *SCO650*9) were identified in this study as dependent on SigB. The *SCO5749* gene encoding the atypical OsaB response regulator was previously identified as partially dependent on SigB under osmotic stress conditions [24]. All other identified SigB-dependent genes represent new members of the SigB regulon in *S. coelicolor* A3(2). Some of them may have a function in osmoprotection, for example a possible transporter for amino acids SCO6014. Furthermore, proteins involved in cell wall biogenesis (SCO0565, SCO2509, SCO3557, SCO5998) may also increase tolerance to osmotic stress. Interestingly, the *rrnE* operon for 16S, 23S, 5S rRNA is also partially controlled by SigB. Its enhanced expression in conditions of osmotic stress may also increase the fitness of the strain in such conditions. All other proteins have no specific function (Table 1).

Similar functions in osmoregulation and the response to oxidative stress have also been described for many of the osmotic stress-induced SigB regulon genes identified by DNA microarray [25]. To our surprise, although we used two overlapping libraries (each containing about 150,000 colonies), of the 47 positive clones identified, only 14 representative SigB-dependent promoters were identified. Although the DNA microarray approach also identified indirect SigB-dependent genes (including larger operons), the difference between the two approaches is relatively high. In addition, nine new target genes were identified using the two-plasmid system. One reason for this small number of identified SigB-dependent genes may be the efficiency of the system. The system can identify direct promoters dependent upon SigB, and it is probable that weaker SigB-dependent promoters cannot be distinguished after screening blue colonies. Thus, only strong SigB-dependent promoters can be identified with this two-plasmid system. Another explanation may be that, in addition to SigB, some SigB-dependent promoters may be regulated by some other transcriptional regulator. In fact, transcription of the previously characterized *dpsAp* promoter, which was dependent upon two SigB homologues, SigB and SigH, in *S. coelicolor* A3(2) under osmotic stress conditions response, was modulated by the WhiB transcriptional regulator [47]. This promoter, after cloning in the reporter plasmid pSB40N, was not active with any of the SigB homologues in the *E. coli* two-plasmid system (data not shown). Therefore, this heterologous *E. coli* two-plasmid system does not recognize SigB-dependent promoters modulated by other transcription factors.

Since SigB is strongly induced under osmotic stress conditions (Figure 1a), to confirm the promoters in vivo, we performed high-resolution S1-nuclease mapping using RNA isolated from the wild-type *S. coelicolor* M145 and the *S. coelicolor* Δ*sigBop* and *S. coelicolor* Δ*sigHop* mutant strains under osmotic stress conditions similar to those described in [25] (Figure 4). Almost all identified SigB-dependent promoters were confirmed in vivo in *S. coelicolor* M145. They were not active in exponentially cultured cells, but their expression was induced under osmotic stress conditions and, under these conditions, were absent or reduced in the *S. coelicolor* Δ*sigBop* mutant. Interestingly, some promoters partially dependent upon SigB were also reduced in the *S. coelicolor* Δ*sigHop* mutant. The previously characterized SigH-dependent *sigHp2* promoter [27] was not affected in *S. coelicolor* Δ*sigBop*, but was neither present in *S. coelicolor* Δ*sigHop*. The constitutive *rrnEp1* promoter identified upstream of the SigB-dependent *rrnEp2* provided relatively constant expression levels using the same RNA samples and can represent a positive quality control for RNA samples (Figure 4). The three identified SigB-dependent promoters (*SCO0566p*, *SCO0279p*, *SCO3557p*) were not identified in vivo in *S. coelicolor* M145 under osmotic stress conditions. They are probably expressed under different conditions. These data suggest that some SigB-dependent promoters are recognized by other SigB homologues. Several identified promoters from the SigB regulon, including *sigB* itself, were also partially transcribed in the *sigB* mutant [25]. Likewise, some other characterized SigB-dependent genes were dependent on other SigB homologues [24,47].

### 2.4. Cross-Recognition of Promoters by Nine Homologues of SigB

Interestingly, the two identified SigB-dependent promoters (*rrnEp2* and *SCO3557p*) were previously identified by similar two-plasmid screening with the sigma factors SigG and SigH, respectively [20,39]. This suggests cross-recognition of promoters by several SigB homologues in *S. coelicolor* A3(2). Therefore, we used this two-plasmid system to analyze the recognition of all identified SigB-dependent promoters and already published SigF-, SigG-, and SigH-dependent promoters [20,36,37,38,39] with all nine SigB homologous sigma factors from *S. coelicolor* A3(2) (SigB, SigF, SigG, SigH, SigI, SigK, SigL, SigM, SigN).

The genes for all nine sigma factors were PCR-amplified from annotated N-terminal methionine and cloned into pAC5mut2 under the control of the *trc* promoter, resulting in pAC-sigB, pAC-sigG, pAC-sigI, pAC-sigK, pAC-sigL, pAC-sigM, pAC-sigN. Sigma factor SigF contains an unusual short repeat N-terminal region [7]. Therefore, in addition to the previously published pAC-sigF1 [36], we prepared a new plasmid pAC-sigF2 containing N-terminally truncated SigF (from MSRGADTR…), which lacked this repeat region. Sigma factor SigH contains an unusually long N-terminal region compared to all other SigB homologues [7]. Therefore, in addition to the previously published pAC-sigH1 [38], we prepared a new plasmid pAC-sigH2 containing the entire annotated SigH, including this N-terminal region. Subsequently, each plasmid containing a sigma factor-dependent promoter was transformed into *E. coli* containing an expression pAC plasmid with a particular sigma factor gene. In contrast to screening conditions to identify the SigB-dependent promoters, we used cheaper MacConkey agar plates, which gave similar results as LB plates with X-gal (Figure 5a). The final results revealed cross-recognition of most promoters by more than one SigB homologue (Figure 5b).

All promoters recognized by the full-length sigma factor SigF (pAC-sigF1) were similarly active with plasmid pAC-sigF2 containing the N-terminally truncated SigF. Therefore, this short N-terminal repeat region probably has no function in SigF activity. However, no promoter recognized by the short form of SigH (pAC-sigH1) was active with plasmid pAC-sigH2 containing the larger form of SigH. These results suggest that this larger form of SigH is inactive or much less active. Therefore, this N-terminal region may have some inhibitory effect on sigma factor function. These results are consistent with the immunodetection analysis (Figure 1b), where this N-terminal region is proteolytically removed during differentiation and the short form of SigH is accumulated in later developmental stages, where a role of SigH is presumed. This proteolysis is therefore a regulatory process for activation of the inactive long form of SigH at later developmental stages.

Almost all 24 examined promoters were recognized by two or more SigB homologues. These results are consistent with phylogenetic analysis of SigB homologues of *S. coelicolor* A3(2) and comparison of their amino acid sequences. All nine SigB homologues were highly similar to *B. subtilis* SigB, mainly in the 2.4 and 4.2 regions, which are involved in recognition of the -10 and -35 promoter regions, respectively. Moreover, all amino acid residues in these regions directly involved in promoter recognition were identical in all SigB homologues, clearly indicating the recognition of very similar promoters [7].

Although there is some variability in the recognition of promoters by these nine SigB homologues, the results suggest some interesting features. Most of the promoters recognized by SigB are also recognized by SigL. These two sigma factors are activated mainly by osmotic stress conditions in the hierarchical order SigB → SigL and are present in the substrate mycelium [25]. Based on phylogenetic analysis and comparison of SigB homologues [7], these two sigma factors are very similar. They contain almost identical 2.4 and 4.2 regions and are closely located in one branch in the phylogenetic tree. Therefore, their cognate promoters should be most similar, which is consistent with the results of cross-recognition. On the other hand, several promoters recognized by SigF are also recognized by SigH and SigN. All these sigma factors play a role in morphological differentiation and are expressed at various developmental stages. SigF is essential for spore maturation and its expression is restricted to sporulating aerial hyphae [10,12]. SigH also has a role in the sporulation of aerial hyphae and is spatially located in sporulating aerial hyphae [27,29]. Similarly, SigN plays a role in morphological differentiation and is expressed in the subapical stem region of aerial hyphae [16]. However, detailed analysis of the promoter sequences for these two groups did not reveal any specific common features of their promoters (Table 2). Likewise, other common promoter recognition groups do not have any specific sequence features that would distinguish them from other groups. All have only high similarity in the -35 and -10 promoter regions, which are similar to the consensus sequence of promoters recognized by homologous *B. subtilis* SigB (GTTTAA–N12-14–GGGA/TAA/T) [44]. All promoters have almost invariant first three residues G and residue A at the fifth position of the -10 region, with less conservation of the other two positions. The -35 region is less conserved. However, there are some small differences between the groups in this region. Promoters recognized by sporulation-specific sigma factors (SigF, SigH, SigN) have a more conserved GTTT motif in this region, compared to another group of promoters recognized by osmotic stress induced sigma factors (SigB, SigL, SigM), which have a less conserved -35 region (GNNT). Interestingly, the promoters recognized by six different SigB homologues (*SCO3557p* and *rrnEp2*) are most similar to the *B. subtilis* SigB consensus sequence. In fact, when we tested the representative *B. subtilis* SigB-dependent promoter *Pctc* (GTTTAA–N14–GGGTAT) [44] in our two-plasmid system, it was active with all nine *S. coelicolor* A3(2) SigB homologues (data now shown). Consistent with these results, in vitro transcriptional analysis with the *Pctc* promoter revealed that several *S. coelicolor* A3(2) SigB homologues were able to transcribe this heterologous promoter. This promoter was activated by osmotic stress conditions in *S. coelicolor* A3(2), but not by other stress conditions. These data also suggest that osmotic stress is controlled by several SigB homologues with overlapping promoter specificity [48]. Therefore, in contrast to the unicellular *B. subtilis* model, where different environmental or energy stresses activate one general stress response alternative sigma factor SigB [6], the situation is more complex in *S. coelicolor* A3(2). Several SigB homologues with overlapping promoter specificities are required to manage osmotic stress, the regulation of which is associated with morphological differentiation [4]. However, the detailed mechanism of this connection is still unclear. Some data suggest that the internal turgor may be an important osmotic signal in this link, because it is present in two developmental stages, which are associated with growth arrest, at the beginning of aerial mycelium formation and during aerial mycelium septation [1,7]. These overlapping specificities may also be associated with specific cell types during development, as several SigB homologues (SigF, SigN) are spatially located in different cell types as described above.

Our data, together with recently published results regarding the interconnection of various anti-sigma factors in the regulation of these SigB homologues [33], suggest a complex regulation of the osmotic stress response in relation to morphological differentiation compared to the *B. subtilis* SigB model [6]. A similar complex interconnected model between these SigB homologues and their regulons was also proposed by computational modelling of DNA microarray data during spore germination of *S. coelicolor* A3(2) [19]. Further comprehensive work is needed to elucidate these overlapping promoter specificities of SigB homologues and their complex activation. Partial clarification in this regard can be obtained after mutagenesis of some promoters to identify specific bases necessary for the recognitions of SigB homologues. In addition, analysis of the expression of these promoters in *S. coelicolor* mutants in all nine sigma factor genes will be required to confirm this cross-recognition. These experiments are in progress.

## 3. Materials and Methods

### 3.1. Bacterial Strains, Culture Conditions, and Plasmids

All strains and plasmids used in this study are described in the Table 3. The conditions for *E. coli* growth and transformation were described in [49]. Luria-Bertani (LB) medium was used for *E. coli* growth. If required, the media were supplemented with 100 μg/mL ampicillin (Amp), 50 μg/mL apramycin (Apr), 50 μg/mL kanamycin (Kan), and 40 μg/mL chloramphenicol. *E. coli* colonies containing sigma factor-dependent promoters using the *E. coli* two-plasmid system [35] were screened on LBACIX plates [LB with Amp, Cm, 1 mM isopropyl-β-D-thiogalactopyranoside (IPTG), and 20 μg/mL X-gal] or MacConkey agar plates (BIOMARK Laboratories, India) with Amp, Cm, and 1 mM IPTG. To isolate RNA from *E. coli* cultures, strains containing the corresponding plasmids of the two-plasmid system were grown in 20 mL of LB medium with Amp, Cm, and 1 mM IPTG at 37 °C to the stationary phase (16 h). Growth and handling of *S. coelicolor* A3(2) were carried out as described in [50]. Minimal solid medium MM + 0.5% mannitol and rich solid media SFM, GYM and R2YE [50] were used to assess morphological differentiation. Minimal NMP liquid medium [50] was used to grow *S. coelicolor* A3(2) strains. To isolate RNA from liquid-grown *S. coelicolor* cultures, 10^9^ colony-forming units (CFU) were inoculated into 50 mL of liquid NMP medium containing 0.5% mannitol as a carbon source [50], and allowed to grow to the late exponential phase (20 h). RNA from osmotic stress conditions was prepared from *S. coelicolor* cells grown to the late exponential phase (20 h) and subjected to 0.5 M NaCl or 1 M sucrose for 30 and 60 min. For immunoblot analysis from cultures grown in liquid medium, 10^9^ CFU of *S. coelicolor* strain were inoculated into 50 mL of liquid NMP medium containing 0.5% mannitol as a carbon source and allowed to grow to the early exponential phase (12 h), the late exponential phase (20 h), and the stationary phase (30 h). Cell-free protein extracts under osmotic stress conditions were prepared from the cells grown to the late exponential phase (20 h) and subjected to 0.5 M NaCl for 30 min. For immunoblot analysis from surface cultures during morphological differentiation, 10^8^ CFU of *S. coelicolor* strain were spread on a cellophane disc placed on the surface of solid agar minimal medium MM and grown to various developmental phases; 1 d—vegetative substrate mycelium, 2 d—beginning of aerial mycelium formation, 3 d—aerial mycelium, 4 d—septation of aerial mycelium, and 5 d—spore maturation.

### 3.2. Recombinant DNA Techniques

Standard DNA manipulation methods were performed as described in [49]. Chromosomal DNA from *S. coelicolor* A3(2) strains was prepared according to [50]. Southern blot hybridization analysis was performed as described in [49]. 1 μg of DNA was digested with restriction enzymes, separated by electrophoresis in a 0.8% (*w*/*v*) agarose gel, and transferred to a Hybond N membrane (Roche, Germany). Hybridization was performed according to the standard DIG protocol (Roche, Germany) using DIG-labelled probes. Signals were detected by DIG chemiluminescent detection kit using CSPD (Roche, Germany). DNA sequencing was carried out with the ABI PRISM^TM^ Dye Terminator Cycle Sequencing Ready Reaction Kit (Applied Biosystems, Norwalk, CT, USA), and analyzed on an automatic DNA sequencer (Applied Biosystems model 373). DNA sequence ladders G, G + A, T + C, C for S1-nucleaase mapping were performed by chemical method [46].

To construct new expression vectors that have sigma factor genes under the control of the *trc* promoter, the genes encoding sigma factors were amplified by: PCR with high fidelity *Pfu* DNA polymerase (Stratagene, La Jolla, CA, USA); *S. coelicolor* M145 chromosomal DNA as a template; selected primers SigXNde and SigXHind (Appendix A) to introduce an *Nde*I site next to the translation initiation codon; and a *Hin*dIII site downstream of the stop codon. The amplified DNA fragments were digested with *Nde*I and *Hin*dIII, and ligated into plasmid pAC5mut2 digested with the same restriction enzymes, resulting in the final recombinant plasmids pAC-sigB, pAC-sigF2, pAC-sigH2, pAC-sigI, pAC-sigK, pAC-sigL, pAC-sigM, pAC-sigN. The nucleotide sequences of all constructs were verified by sequencing. The plasmids pAC-sigF1, pAC-sigH1, pAC-sigG1 have been described previously [20,36,38].

### 3.3. Detection of E. coli Clones Containing SigB-Dependent Promoters and Cross-Recognition of Promoters by Nice SigB Homologous Sigma Factors

Two *S. coelicolor* M145 libraries with small DNA fragments (0.5–1 kb *Taq*I cloned into the *Cla*I site of pSB40N and *Sau*3AI cloned into the *Bam*HI site of pSB40N) [36] were used to transform *E. coli* XL1-Blue strain containing compatible plasmid pAC-sigB. Clones containing SigB-dependent positive promoters were screened on LBACIX plates according to the procedure described in [35]. The procedure is illustrated in Figure 2.

To examine the cross-recognition of promoters by nine SigB homologous sigma factors (SigB, SigF, SigG, SigH, SigI, SigK, SigL, SigM, SigN) from *S. coelicolor* A3(2), a plasmid containing a promoter DNA fragment (cloned in pBS40N) was transformed into the *E. coli* XL1-Blue strain containing a compatible expression plasmid having a sigma factor gene under the control of the *trc* promoter (pAC-sigB, pAC-sigF1, pAC-sigF2, pAC-sigG, pAC-sigH1, pAC-sigH2, pAC-sigI, pAC-sigK, pAC-sigL, pAC-sigM, pAC-sigN). Transformation mixtures were screened on MacConkey plates with Amp, Cm, and 1 mM IPTG.

### 3.4. RNA Isolation and S1-Nuclease Mapping

Total RNA was isolated as described in [55]. Its integrity was indicated as sharp bands of 23S rRNA and 16S rRNA after agarose gel electrophoresis containing 2.2 M of formaldehyde. High-resolution S1-nuclease mapping of promoter TSS was performed as described in [55]. 40 μg of each RNA was hybridized with 0.02 pmol of an appropriate DNA probe labelled at one 5′ end with [γ-^32^P] ATP and treated with 120 U of S1-nuclease (Biolabs, San Diego, CA, USA). S1 probes for TSS detection in the *E. coli* two-plasmid system were prepared by PCR amplification from a particular identified plasmid containing a SigB-dependent promoter using the 5′ end-labelled universal oligonucleotide primer 47 from the *lacZ*α gene and the primer mut80 from the 5′ region adjacent to the cloning polylinker of pSB40N (Appendix A). S1 probes for detection of TSS in *S. coelocolor* A3(2) were similarly prepared by PCR amplification from a particular identified plasmid containing a SigB-dependent promoter using a reverse 5′ end-labelled specific SCO primer (Appendix A) internally, for a particular *SCO* gene that was driven by the promoter (Figure 3) and the primer mut80 from the 5′ region flanking the cloning polylinker of pSB40N. The S1 probe for the *sigHp2* promoter has been described previously [27]. Oligonucleotides were labelled at their 5′ ends with [γ-^32^P] ATP (ICN, 4500 Ci/mmol) and T4 polynucleotide kinase (Biolabs, San Diego, CA, USA) as described in [49]. Protected DNA fragments were analyzed on DNA sequencing gels (6% polyacrylamide containing 8 M urea) along with G, G + A, T + C, C sequencing ladders from a particular end-labelled DNA fragment [46].

### 3.5. Isolation of SigB, SigH, UshX Proteins and Immunoblot Analysis

Overexpression and purification of proteins used to prepare polyclonal antibodies have been described previously. Purified His-tagged SigB [24] and His-tagged SigH and UshX [31] were analyzed by a 12.5% sodium dodecyl sulfate-polyacrylamide gel electrophoresis (SDS-PAGE) [56]. In all cases, a single band with the mobility described previously was identified. About 1 mg of each protein was used to immunize a 6-month-old New Zealand white rabbit by a standard procedure.

Crude protein extracts from *Streptomyces* strains for immunoblot analysis were prepared as follows: mycelium from liquid-grown cultures was harvested by centrifugation at 12,000× *g* for 10 min and washed with ice-cold 0.9% (*w*/*v*) NaCl. The surface-grown mycelium was scraped from cellophane placed on agar minimal MM medium. All steps were carried out at 4 °C. The mycelium was disrupted by grinding with purified acid-washed sea sand in the presence of liquid nitrogen in a mortar for about 5 min. The mixture was then suspended in binding buffer (12.5 mM Tris-Cl pH 8, 60 mM KCl, 1 mM EDTA, 1 mM DTT, 12% glycerol, 1 mM PMSF) with 1 tablet of protease inhibitors Complete^TM^ (Roche)/10 mL and cell debris was removed by centrifugation for 30 min at 30,000× *g*. Cell-free extracts were stored in aliquots at −80 °C. Protein concentrations were determined according to [57] with bovine serum albumin as a standard.

Crude protein extracts (30 μg) were separated on a 12.5% SDS-PAGE. After electrophoresis, the gel was washed five times with water, incubated in transfer buffer (15 mM Tris, 120 mM glycine, 20% methanol) for 5 min, and the proteins were blotted onto a nitrocellulose ECL membrane (Amersham) in transfer buffer. The membrane was incubated in TBST-200 buffer [50 mM Tris-Cl pH 7.5, 0.2 M NaCl, 0.3% (*v*/*v*) Tween 20] for 15 min at room temperature, then similarly for 15 min in TBST-200* buffer [50 mM Tris-Cl pH 7.5, 0.2 M NaCl, 0.05% (*v*/*v*) Tween 20], and then overnight in blocking buffer [TBST-200* containing 5% (*w*/*v*) skim milk powder] at 4 °C. Polyclonal antibodies were added to blocking buffer (SigB at 1: 2000; SigH at 1: 1000; UshX at 1:1000) and incubated with the membrane for 1.5 h at room temperature. Subsequently, unbound antibodies were removed by two 5-min washes in water, two 5-min washes in TBST-500 buffer [20 mM Tris-Cl pH 7.5, 0.5 M NaCl, 0.05% (*v*/*v*) Tween 20, 5% (*w*/*v*) skim milk powder] and a 5-min wash in CBST-500 buffer [20 mM Na-citrate pH 5.5, 0.5 M NaCl, 0.05% (*v*/*v*) Tween 20, 5% (*w*/*v*) skim milk powder]. The membrane was incubated in CBST-500 buffer containing horseradish peroxidase-conjugated porcine anti-rabbit IgG (Roche) at a dilution 1:10,000 for 1.5 h and then washed for 5 min in water, 5 min in CBST-500 buffer, 2 × 5 min in TBST-500 buffer, and 5 min in TBS-500 buffer [20 mM Tris-Cl pH 7.5, 0.5 M NaCl, 5% (*w*/*v*) skim milk powder]. Reactive bands were visualized by chemiluminescence (ECL plus, Amersham, UK).

### 3.6. Construction of Deletion Mutants in S. coelicolor A3(2)

Construction of the mutant with the deleted operon *rsbB*, *rsbA*, *sigB* in strains *S. coelicolor* M145 was performed by the REDIRECT procedure [41]. The disruption cassette containing *oriT* and the apramycin-resistance gene was prepared by PCR with primers SigBopDir and SigBopRev (Appendix A) from the upstream and downstream regions of the operon using a gel-purified 1384-bp *EcoR*I-*Hind*III fragment from pIJ773 as a template. After purification on a Wizard column (Promega, Madison, WI, USA), 1 μg of the amplified DNA fragment was electroporated into *E.* coli BW25113 containing plasmid pIJ790 and *sigB*-containing SuperCos1-derived cosmid Stf55 [54]. The resulting mutant cosmid Stf55ΔSigBop was transformed into *E.* coli ET12567 containing plasmid pUZ8002, and the *sigBop*-deleted allele was transferred by conjugation to *S.* coelicolor M145. Colonies were screened for Apr resistance and Kan sensitivity, indicating a double crossover. Four such colonies were identified, which resulted in mutant strains *S. coelicolor* Δ*sigBop*-A, B, C, and D and were confirmed by Southern blot hybridization analysis. Hybridization followed the standard DIG protocol (Roche) using DIG-labelled probes A, B, and C (Appendix A). Probe A was a 746-bp *Not*I-*Xho*I DNA fragment covering the *SCO0601* gene, probe B was an 833-bp *Bsp*HI-*Sph*I DNA fragment covering the *sigB* gene, and probe C was a 573-bp *Aat*II-*Pvu*I DNA fragment covering the *rsbA* gene. A similar REDIRECT procedure was used to delete *sigB* alone, except that the disruption cassette was prepared by PCR with primers SigBDir and SigBopRev (Appendix A) from the upstream and downstream regions of the *sigB* gene.

The entire operon *ushY*, *ushX*, *sigH* was deleted by another strategy in the wild-type *S. coelicolor* M145 strain. The Apr-resistant cassette flanked by a strong *T4* terminator from plasmid pHP45Ώaac [53] was cloned as an 1,7-kb *Hin*dIII DNA fragment into pBluescript II SK digested with the same enzyme, resulting in pAAC4A. A 4428-bp *Bsr*GI-*Bsp*HI (blunt-ended with Klenow enzyme) DNA fragment from the upstream region of the *sigH* operon was cloned into pAAC4A digested with *Acc*65I and *Cla*I (blunt-ended with Klenow enzyme), resulting in pSigHdel1. A 2500-bp *Pml*I-*Hin*dIII DNA fragment from pSIGSC1 [28] (covering the downstream region of the *sigH* operon) was cloned into pAPH3 [52] digested with *Eco*RV and *Hin*dIII, resulting in pSigHdel2. Subsequently, a 3800-bp *Eco*RI-*Xba*I fragment from pSigHdel2 was cloned into pSigHdel1 digested with the same enzymes. The resulting plasmid pSigHdel3 was used to transform *S. coelicolor* M145 protoplasts [50]. Colonies were screened for Apr resistance and Kan sensitivity, indicating a double crossover. One such colony was found, resulting in a mutant strain *S. coelicolor* Δ*sigHop*, which was confirmed by Southern blot hybridization analysis. Hybridization followed the standard DIG protocol (Roche) using a DIG-labelled probe from a 1000-bp *Acc*65I-*Bam*HI DNA fragment covering the *SCO5241* gene (Appendix A).

## Figures and Tables

**Figure 1 ijms-22-07849-f001:**
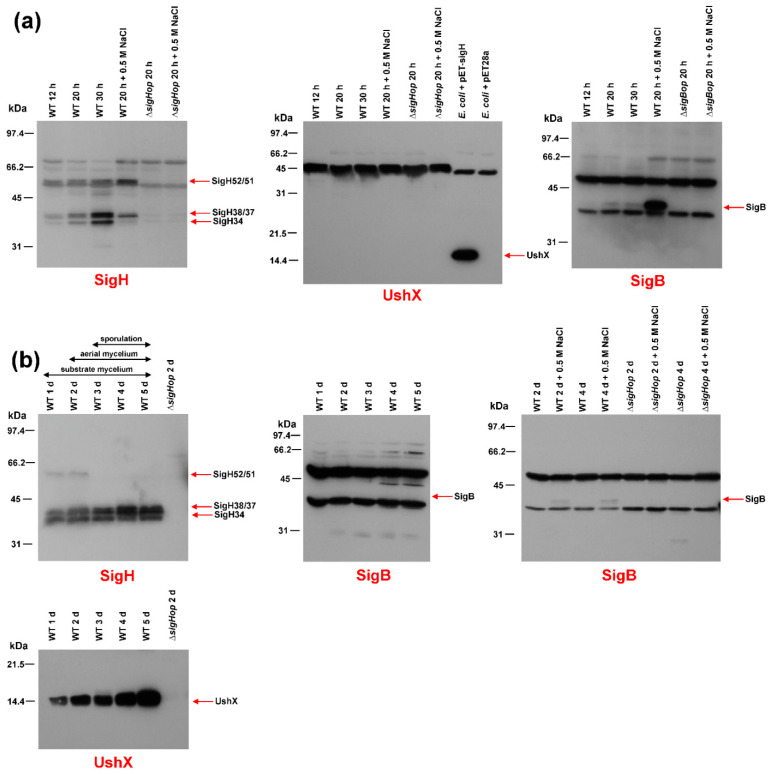
Western blot immunodetection of SigB, SigH and UshX. Crude protein extracts (30 μg) prepared from the wild-type *S. coelicolor* M145 strain (WT) or *S. coelicolor* Δ*sigBop* or Δ*sigHop* mutant strains were analyzed on a 12.5% SDS-PAGE, transferred to a nitrocellulose membrane and detected with the antibody indicated under each blot. (**a**) Crude protein extracts were prepared from the above strain, which was cultured in NMP liquid minimal medium for 12 h (early exponential phase), 20 h (late exponential phase), and 30 h (stationary phase). The crude extracts under osmotic stress conditions were prepared from cultures grown for 20 h (exponential phase) and incubated for 30 min in the presence of 0.5 M NaCl. The *E. coli* BL21(DE3) strain with the plasmid pET-sigH or pET28a was grown to exponential phase. (**b**) Crude protein extracts were prepared from the indicated strain grown on cellophane discs placed on the surface of solid agar minimal medium MM for 1 day (vegetative substrate mycelium), 2 days (beginning of aerial mycelium formation), 3 days (aerial mycelium), 4 days (septation of aerial mycelium), and 5 days (spore maturation). Crude extracts under osmotic stress conditions were prepared from the cultures similarly grown to indicated time points in solid minimal MM medium with 0.5 M NaCl. The position of the specific band for each protein is indicated by an arrow. The positions of the bands in the protein standard are shown on the left.

**Figure 2 ijms-22-07849-f002:**
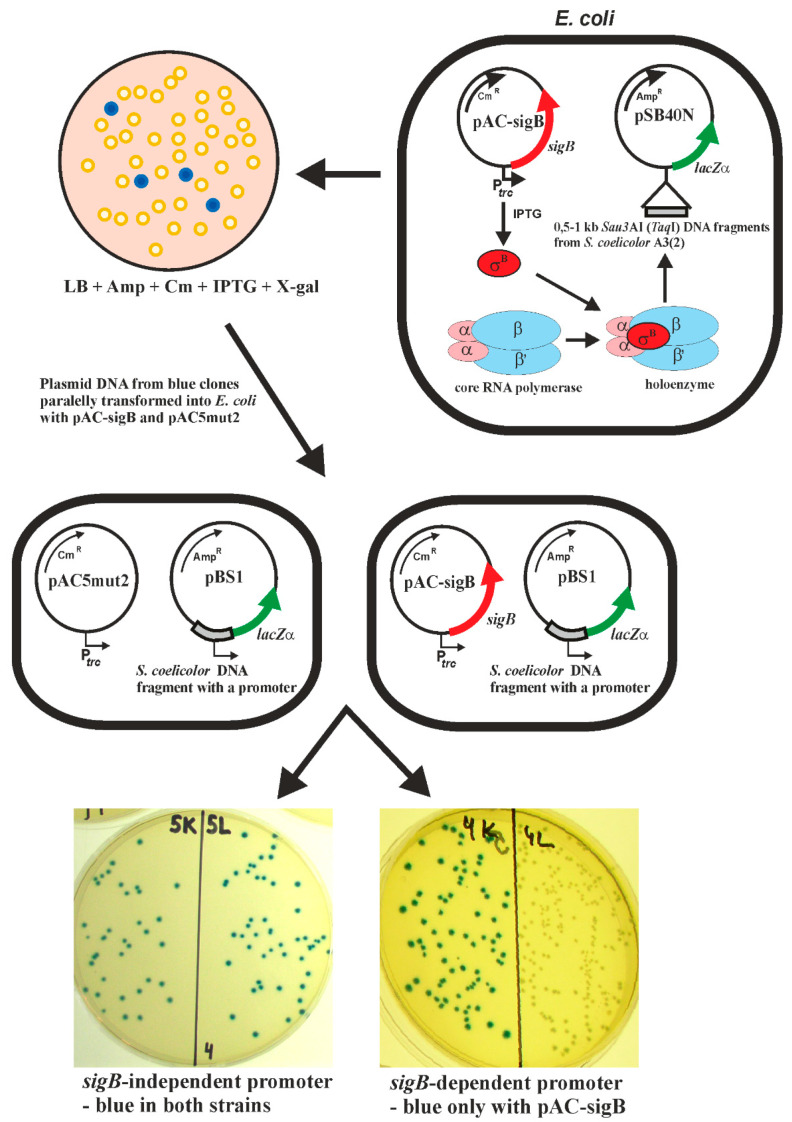
Scheme illustrating the *E. coli* two-plasmid system for identifying sigma factor-dependent promoters (in this case SigB). A library of *Sau3*AI or *Taq*I DNA fragments from *S. coelicolor* A3(2) in pSB40N [36] was transformed into *E. coli* with the expression plasmid pAC-sigB containing the sigB gene under the control of the inducible *trc* promoter and screened on X-gal selective plates. Plasmid DNA from the blue colonies was subsequently transformed into two *E. coli* strains; one strain with the expression plasmid alone (pAC5mut2), and the other with pAC-sigB. The SigB-independent promoter is active in both strains, and SigB-dependent only in *E. coli* with pAC-sigB. The final plates with *E. coli* containing a *sigB*-independent or *sigB*-dependent promoter are shown.

**Figure 3 ijms-22-07849-f003:**
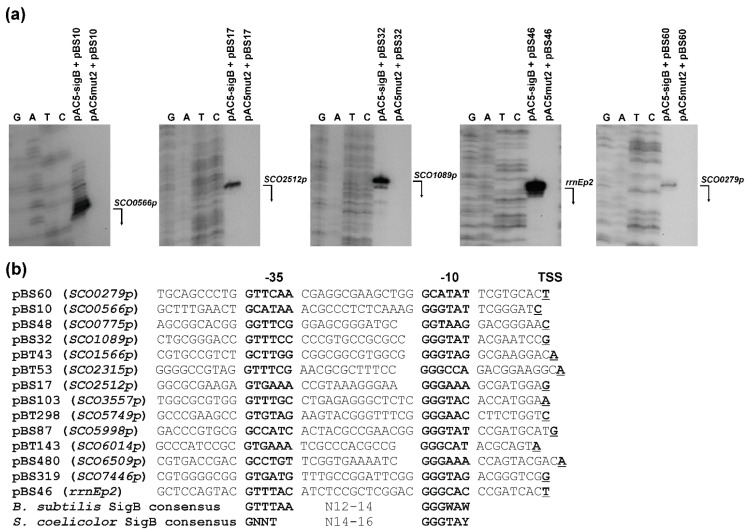
Representative plasmids containing SigB-dependent promoters from *S. coelicolor* A3(2). (**a**) Examples of high-resolution S1-nuclease mapping of transcription start site (TSS) of SigB-dependent promoters in the *E. coli* two-plasmid system. Isolation of RNA and S1-nuclease mapping are described in Material and Methods. The 5′-labeled DNA probes were hybridized with 40 μg of RNA isolated from the stationary phase cultures of *E. coli* containing indicated plasmids and treated with 120 U of S1-nuclease. The RNA-protected DNA fragments were analyzed on DNA sequencing gels together with G, G + A (lane A), T + C (lane T), C sequencing ladders derived from the end-labelled DNA fragments [46]. The bent arrows indicate the positions of the protected fragments. Before assigning the TSS, 1.5 nt was subtracted from the length of the protected DNA fragment to account for the difference in the 3′ ends resulting from S-nuclease digestion and chemical sequencing reactions. (**b**) Comparison of identified SigB-dependent promoters located in representative plasmids with the consensus sequence of promoters recognized by *B. subtilis* SigB [44] and *S. coelicolor* A3(2) SigB [45]. The promoter names are derived from the annotated *SCO* genes from *S. coelicolor* A3(2) (GenBank Acc. No. AL645882), which are under the control of the promoters. The proposed -35 and -10 promoter regions are in bold. The TSS is in bold and underlined. W = A/T, Y = C/T.

**Figure 4 ijms-22-07849-f004:**
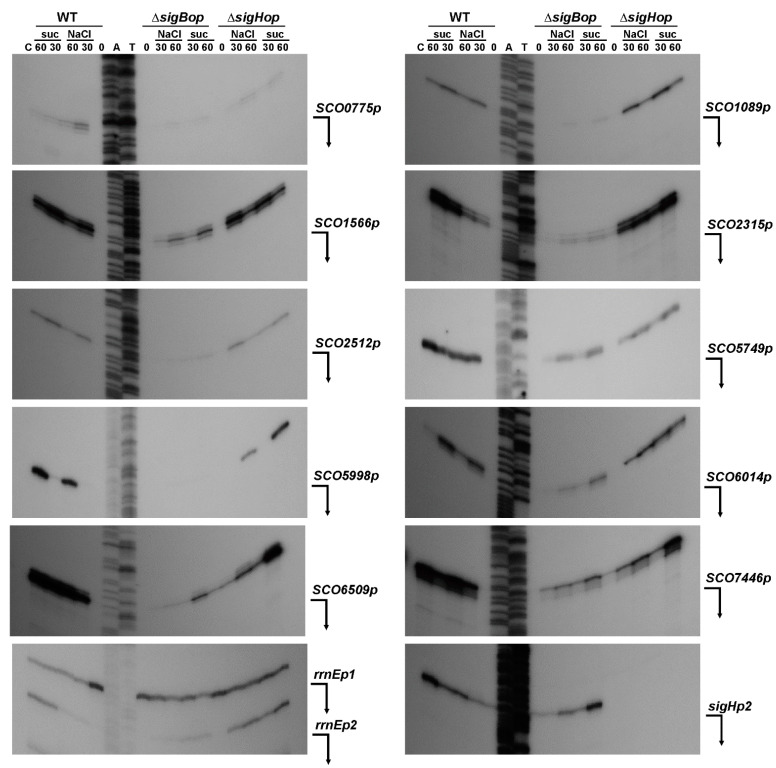
High-resolution S1-nuclease mapping of the TSS for the SigB-dependent promoters in *S. coelicolor* A3(2). The 5′-labeled DNA probes were hybridized with 40 μg of RNA and treated with 120 U of S1-nuclease as described in the Materials and Methods. RNA was isolated from liquid cultured *S. coelicolor* M145 (WT) and its isogenic Δ*sigBop* and Δ*sigHop* mutants. The strains grew into the exponential phase (lane 0), then were exposed to osmotic stress conditions with 0.5 M NaCl and 1 M sucrose (suc) and RNA isolated after 30 or 60 min (lanes 30 or 60). *E. coli* tRNA was used as a control (lane C). Control S1-nuclease mapping was performed with the same RNA samples and a DNA probe for SigH-dependent *sigHp2* promoter [27]. The RNA-protected DNA fragments were analyzed on DNA sequencing gels together with G + A (lane A), T + C (lane T) sequencing ladders derived from the end-labelled DNA fragments [46]. The bent arrows indicate positions of the protected fragments.

**Figure 5 ijms-22-07849-f005:**
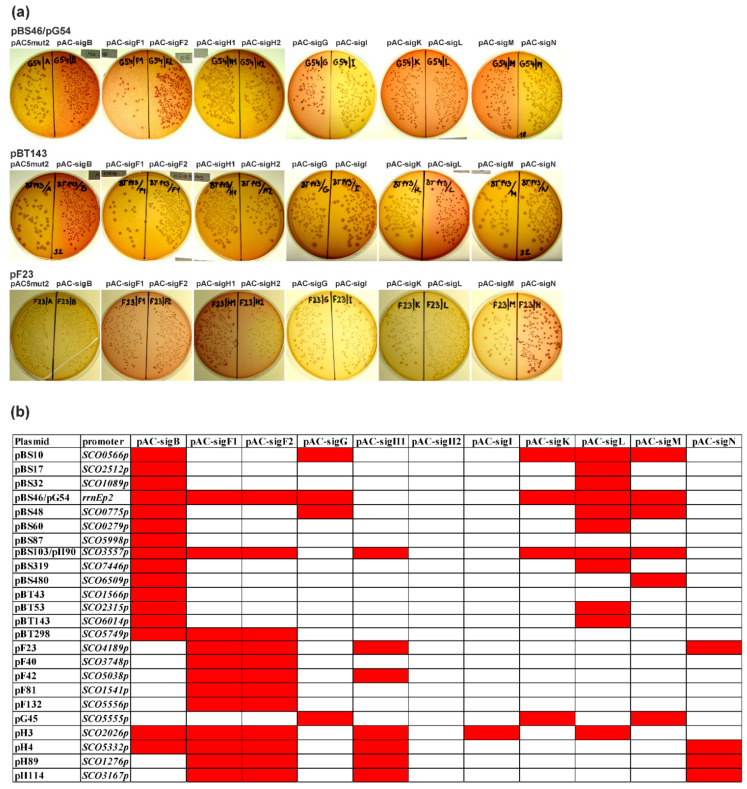
Cross-recognition of all identified SigB-dependent promoters and already published SigF-, SigG-, SigH-dependent promoters [20,36,37,38,39] with all nine SigB homologous sigma factors from *S. coelicolor* A3(2) with the *E. coli* two-plasmid system. (**a**) Examples of MacConkey agar plates (with Amp, Cm) with transformation mixtures of three plasmids containing sigma factor-dependent promoters (pBS46/pG54, pBT143, pF23) with combinations of expression plasmid pAC5mut2 and the vector with cloned sigma factor genes under the control of the *trc* promoter (pAC-). Red colonies correspond to positive results and uncolored colonies to negative results. (**b**) Final results of the cross-recognition analysis with a red field corresponding to positive results and a white field to negative results. The names of the plasmids originally identified as dependent on SigB, SigF, SigG, and SigH are pB, pF, pG, and pH.

**Table 1 ijms-22-07849-t001:** List of identified SigB-dependent genes from *S. coelicolor* A3(2) and their proposed function.

*SCO* Gene or Operon Directed by Identified Promoter	Encoded Protein/Function
*SCO0279*	putative glycosyl hydrolase
*SCO0566*, *SCO0565*, *SCO0564*, *SCO0563*	putative membrane protein, polyprenyl synthase, HP, HP
*SCO0775*	HP
*SCO1089*, *SCO1088*, *SCO1087*, *SCO1086*	HP, putative oxidoreductase, L-treonine aldolase, HP
*SCO1566*	putative acetyltransferase
*SCO2315*	putative membrane protein
*SCO2512*, *SCO2511*, *SCO2510*, *SCO2509*	HP, HP, HP, undecaprenyl phosphate synthase
*SCO3557*, *SCO3556*, *SCO3555*, *SCO3554*, *SCO3553*, *SCO3552*, *SCO3551*	putative septum site determining protein, all other—membrane HP
*SCO5749*	atypical response regulator
*SCO5998*	putative UDP-N-acetylglucosamine transferase
*SCO6014*	putative cationic amino acid transporter
*SCO6509*	hydrophobic protein
*SCO7446*	putative regulator
*rrnE*	16S, 23S, 5S rRNA operon

HP, Hypothetical protein.

**Table 2 ijms-22-07849-t002:** Comparison of promoters dependent on several groups of SigB homologous sigma factors from *S. coelicolor* A3(2).

Plasmid	Promoter	Sequence	35 Region	10 Region	TSS
**SigB-dependent**					
**pBS87**	***SCO5998p***	GACCCGTGCG	**GCCATC**	ACTACGCCGAACGG **GGGTAT**	CCGATGCAT**G**
**pBT43**	***SCO1566p***	CGTGCCGTCT	**GCTTGG**	CGGCGGCGTGGCG **GGGTAG**	GCGAAGGAC**A**
**SigBL-dependent**					
**pBS17**	***SCO2512p***	GGCGCGAAGA	**GTGAAA**	CCGTAAAGGGAA **GGGAAA**	GCGATGGA**G**
**pBS32**	***SCO1089p***	CTGCGGGACC	**GTTTCC**	CCCGTGCCGCGCC **GGGTAT**	ACGAATCC**G**
**pBS60**	***SCO0279p***	TGCAGCCCTG	**GTTCAA**	CGAGGCGAAGCTGG **GCATAT**	TCGTGCAC**T**
**pBS319**	***SCO7446p***	CGTGGGGCGG	**GTGATG**	TTTGCCGGATTCGG **GGGTAG**	ACGGGTCG**G**
**pBT53**	***SCO2315p***	GGGGCCGTAG	**GTTTCG**	AACGCGCTTTCC **GGGCCA**	GACGGAAGGC**A**
**pBT143**	***SCO6014p***	GCCCATCCGC	**GTGAAA**	TCGCCCACGCCG **GGGCAT**	ACGCAGT**A**
**SigBM-dependent**					
**pBS480**	***SCO6509p***	CGTGACCGAC	**GCCTGT**	TCGGTGAAAATC **GGGAAA**	CCAGTACGAC**A**
**SigBF-dependent**					
**pBT298**	***SCO5749p***	GCCCGAAGCC	**GTGTAG**	AAGTACGGGTTTCG **GGGAAC**	CTTCTGGT**C**
**SigBGLM-dependent**					
**pBS48**	***SCO0775p***	AGCGGCACGG	**GGTTCG**	GGAGCGGGATGC **GGTAAG**	GACGGGAA**C**
**SigBGKLM-dependent**					
**pBS10**	***SCO0566p***	GCTTTGAACT	**GCATAA**	ACGCCCTCTCAAAG **GGGTAT**	TCGGGAT**C**
**SigBFGKLM-dependent**					
**pBS46/pG54**	***rrnEp2***	GCTCCAGTAC	**GTTTAC**	ATCTCCGCTCGGAC **GGGCAC**	CCGATCAC**T**
**SigBFHKLM-dependent**					
**pBS103/pH90**	***SCO3557p***	TGGCGCGTGG	**GTTTGC**	CTGAGAGGGCTCTC **GGGTAC**	ACCATGGA**A**
**SigBFHIL-dependent**					
**pH3**	***SCO2026p***	TTACTCGCTG	**GTTTCG**	AGCGCGTAATCTCC **GGGCAT**	GAGCCGT**A**
**SigBFHN-dependent**					
**pH4**	***SCO5332p***	GGCGAGGTAC	**GTTTGC**	GAAGAAGGTGGCGT **GGGCAA**	GATCCGC**C**
**SigF-dependent**					
**pF40p**	***SCO3748p***	CCGGGTTGTT	**GATTCA**	TTTCGTGCTAGGCT **GGACAT**	CA**G**
**pF81**	***SCO1541p***	ACTCAGAGGG	**GTTTAC**	AACGGCACCGTAGG **TGGCAT**	GTCGATT**T**
**pF132**	***SCO5556p***	TGATTAGCCC	**GTTTCG**	TGTTTGATTTGC **GGTATA**	TATCTGC**C**
**SigFH-dependent**					
**pF42**	***SCO5038p***	TACGAGCGGG	**GTTCAA**	GGTTGTGATCCGGT **GGTAAA**	GTGAT**A**
**SigFHN-dependent**					
**pF23p**	***SCO4189p***	TTTGAAACTT	**GTTTAT**	AGGTAGTAGCCAAGT **GGGGAC**	AACATA**G**
**pH89**	***SCO1276p***	GCGTATTCCG	**GTTTGC**	CGCTCCAGGGCCTC **GGGTAG**	AAAATCCAC**A**
**pHT114**	***SCO3167p***	GAACGAAACC	**GTTTCG**	TTTCGCTAGGCGCTG **GGGTAC**	ATTCGTC**A**
**SigGKM-dependent**					
**pG45**	***SCO5555p***	AACTTCCCAC	**GTTAGA**	GGCGTGGTTGCCG **GGGTAC**	TCAAGGCAC**T**

**Table 3 ijms-22-07849-t003:** Bacterial strains and plasmids used in this study.

Strain or Plasmid	Genotypes and Relevant Characteristics ^a^	Reference or Source
Strains		
*E. coli* DH5α	F^−^ *supE44* Δ*lacU169* (ϕ80d*lacZψ*ΔM15) *hsdR17 recA1 endA1 gyrA96 thi-1 relA1* Host strain for plasmid cloning and propagation	Invitrogen
*E. coli* XL1-Blue	*recA1 endA1 gyrA96 thi-1 hsdR17 supE44 relA1 lac* [F′ *proAB lacI*^q^*Z*Δ*M15 Tn10* (Tet^r^)]. Host strain for two-plasmid system	Stratagene
*E. coli* BW25113/pIJ790	Δ(*araD-araB*)*567* Δ*lacZ4787*(::*rrnB-4*) *lacIp-4000* (lacI^Q^) $\bar{\lambda }$ *rpoS369*(Am) *rph-1* Δ(*rhaD-rhaB*) *hsdR514*. Host strain containing pIJ790 for REDIRECT strategy	[41]
*E. coli* ET12567/pUZ8002	*dam-13*::Tn*9 dcm-6 hsdM.* A methylation-deficient host strain containing non-transmissible pUZ80023 for conjugation	[50]
*S. coelicolor* M145	wild-type, prototrophic SCP1^−^ SCP2^−^ Pgl^+^	[50]
*S. coelicolor* Δ*sigB*	Δ*sigB::aac3(IV*), Apr^R^, deleted *sigB* gene in *S. coelicolor* M145 and replaced with the *aac3(IV)* gene from pIJ773	this study
*S. coelicolor* Δ*sigBop*	Δ(*rsbB rsbA sigB*)*::aac3(IV*), Apr^R^, deleted *rsbB*, *rsbA*, *sigB* operon in *S. coelicolor* M145 and replaced with the *aac3(IV)* gene from pIJ773	this study
*S. coelicolor* Δ*sigHop*	Δ(*ushY ushX sigH*)*::aac3(IV*), Apr^R^, deleted *ushY*, *ushX*, *sigH* operon in *S. coelicolor* M145 and replaced with the *aac3(IV)* gene from pHP45 aac	this study
Plasmids		
pBluescript II SK	Amp^R^, standard *E. coli* cloning vector	Stratagene
LITMUS 28	Amp^R^, standard *E. coli* cloning vector	New England Biolabs
pHP45	Amp^R^, standard *E. coli* cloning vector	[51]
pAPH3	Amp^R^, Kan^R^, LITMUS 28 derivative containing the *neo* gene of Tn*5*	[52]
pHP45 aac	Amp^R^, Apr^R^, pHP45 derivative containing the *aacC4* gene	[53]
pAAC4A	Amp^R^, Apr^R^, pBluescript II SK derivative containing the *aacC4* gene	this study
pSIGSC1	Amp^R^, pBR322 derivative containing the *ushY*, *ushX*, *sigH* operon	[28]
pSigHdel1	Amp^R^, Apr^R^, pAAC4A derivative containing the *sigH* upstream region	this study
pSigHdel2	Amp^R^, Kan^R^, pAPH3 derivative containing the *sigH* downstream region	this study
pSigHdel3	Amp^R^, Kan^R^, Apr^R^, pAPH3 derivative containing deleted *sigH* operon and replaced with the *aacC4* gene	this study
pIJ773	Amp^R^, Apr^R^, pBluescript KS derivative containing the *aac(3)IV* gene and *oriT*	[39]
Stf55	Amp^R^, Kan^R^, SuperCos-1 derivative containing the *sigH* operon	[54]
Stf55DSigBop	Amp^R^, Kan^R^, Apr^R^, Stf55 derivative containing deleted *sigH* operon and replaced with the *aac3(IV)* gene	this study
Stf55DSigB	Amp^R^, Kan^R^, Apr^R^, Stf55 derivative containing deleted *sigH* gene and replaced with the *aac3(IV)* gene	this study
pAC5mut2	Cm^R^, expression plasmid vector with the *trc* promoter	[35]
pAC5-sigF1	Cm^R^, pAC5mut2 containing the *sigF* gene under the *trc* promoter	[36]
pAC5-sigG1	Cm^R^, pAC5mut2 containing the *sigF* gene under the *trc* promoter	[20]
pAC5-sigH1	Cm^R^, pAC5mut2 containing the *sigF* gene under the *trc* promoter	[38]
pAC5-sigB	Cm^R^, pAC5mut2 containing the *sigB* gene under the *trc* promoter	this study
pAC5-sigI	Cm^R^, pAC5mut2 containing the *sigI* gene under the *trc* promoter	this study
pAC5-sigK	Cm^R^, pAC5mut2 containing the *sigK* gene under the *trc* promoter	this study
pAC5-sigL	Cm^R^, pAC5mut2 containing the *sigL* gene under the *trc* promoter	this study
pAC5-sigM	Cm^R^, pAC5mut2 containing the *sigM* gene under the *trc* promoter	this study
pAC5-sigN	Cm^R^, pAC5mut2 containing the *sigN* gene under the *trc* promoter	this study
pAC5-sigF2	Cm^R^, pAC5mut2 containing the N-terminally truncated *sigF* gene under the *trc* promoter	this study
pAC5-sigH2	Cm^R^, pAC5mut2 containing the *sigH* gene including longer N-terminal region under the *trc* promoter	this study
pSB40N	Amp^R^, promoter probe plasmid containing a promoterless *lacZ*α gene	[36]

^a^ Cm^R^, chloramphenicol resistance; Amp^R^, ampicillin resistance; Apr^R^, apramycin resistance; Kan^R^, kanamycin resistance.

## Data Availability

Data are available on request.

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
