# Peer review of "Cross-Recognition of Promoters by the Nine SigB Homologues Present in *Streptomyces coelicolor* A3(2)"

_ijms, 2021, doi:10.3390/ijms22157849_

Round 1
Reviewer 1 Report
The paper is clearly written and provides useful information about SigB-type sigma factors in Streptomyces coelicolor that extends and complements previous work by this and other research groups. The work is well documented and appears technically sound. I have only minor suggestions and comments.
- It is stated that the colony phenotype of the DELTAsigB mutant was similar to the DELTAsigBop mutant and that none of the mutants were affected in morphological differentiation. However, the sigB mutant appears subtly different from the other two strains in several panels in Fig. 1, both in the M145 and J1501 backgrounds (e.g. slightly paler on SFM at 10 days in both backgrounds). Perhaps authors could clarify this.
- Also the DELTAsigHm mutant appears slightly different from the DELTAsigHop mutant in colony appearance on SFM and other media in Fig. 4 (e.g. upper left panel). Why is it different?
- On pages 6-9 conclusions are drawn regarding morphological differentiation, but data are only presented on colony appearance. In the absence of any microscopical examination, the conclusions should be phrased a bit more carefully.
- On page 15, it would be interesting to know how many clones were fished out that had promoters that were constitutively active in E. coli. This is not clearly stated. I would have guessed that a substantial number of promoters could show constitutive activity in E. coli, independently of the expressed SigB-type sigma factor.
- The text is at places unnecessarily long and wordy. Particularly, the Discussion is very long (4 full and densely written pages) and contains a lot of re-iteration of the results. It must be possible to condense the presentation and particularly the discussion.
- In the conclusion in line 257, it says that deletion of sigH operon “had no effect on … osmotic response”. Note that the data that this conclusion is based on are only visible phenotypic effects seen as colony appearance and growth on agar plates. It can not be excluded that the sigH operon affects expression of genes in response to osmotic stress response, even though this does not visibly affect the phenotype. Or do I misunderstand what the authors are trying to say?
Some minor suggestions:
Fig. 5: Point out UshX in panel A with arrow. It would also be good to comment on the prominent non-specific (?) bands in this and some of the other blots.
Delete first sentence of Introduction (comes from the manuscript template?)
How large were the genomic libraries used in screening for promoters? Is it likely that they represent the entire genome?
Table 1: Spell out in footnote what HP means. Hypothetical protein?
Lines 497-499: Not clear what the authors are trying to say here. Perhaps something like “did not shed light on….”?
Table 3: The genotypes for the new sigB, sigBop, and sigHop mutants should be spelled out in a proper way. May for example be DELTA(rsbB rsbA sigB)::apra, and with some explanation of the exact content of the “apra cassette”.
Reviewer 2 Report
In their manuscript entitled “Cross-Recognition of Promoters by Nine Homologues of Alternative Stress Response Sigma Factor SigB in Streptomyces coelicolor A3(2)” the authors try to decipher the relationships existing between the 9 SigB-like sigma factors of Streptomyces coelicolor A3. They first demonstrated that SigB is mostly involved in the response to osmotic stress and not in the differentiation process as initially thought even if the two processes seems to be linked. The authors also described the identification of novel SigB-dependent promoters using a two plasmids system in the heterologous host E. coli. In this system, one of the plasmid expresses the sigma factor of interest and the other plasmid carries small fragments of the S. coelicolor genome fused to reporter gene LacZ. Colonies carrying plasmids containing fragments with promoters recognized by a given sigma factor express the LacZ gene and are blue in presence of X-gal or red on MacConkey medium. Using this E. coli two-plasmids system the authors also tested the ability of the 9 SigB-like factors (and two truncated derivatives of SigF and SigH) to transcribe these novel SigB dependent promoters as well some SigF, SigG and SigH dependent promoters, previously characterized. This study demonstrated the cross recognition of the different promoters by more than one and up to five sigma factors, at least in the heterologous host, E. coli. However, the authors could not establish a clear consensus sequence specific for each sigma factor; perhaps because of the limited number of promoter fragments available. Nonetheless, such specificity is predicted to exist, at least for SigI that recognizes a single promoter (SCO2026), but might be modulated by other transcriptional regulators.
The paper is interesting and includes an impressive amount of highly skilled experimental work even if it is, by some aspects, mainly confirmatory of previously work. It has also some weaknesses that could be pointed at. For instance, all comments on the impact of deletion of sigma factors on growth in conditions of low or high osmotic stress or on differentiation (sporulation and antibiotic production) are not supported by quantitative data such as growth curves, antibiotic assays etc..(Figure 2 and 4). The authors are only providing picture of plates but this is inaccurate and could be misleading and none of the constructed sigma mutants were complemented. Furthermore key information concerning the temporal expression of the sigma factors is only provided for SigB and SigH (Figure 5). In which temporal order are the other sigma factors expressed? Are they all expressed concomitantly or sequentially? If possible, a scheme summarizing what is known about the temporal expression, the regulation of these 9 sigma factors, especially by osmotic stress and on their regulatory links (which transcribes which?) would be most welcome.
At last, the paper is far too long because of numerous repetitions. To reduce the length of this paper, it is recommended to condense the two sections “Results” and “Discussion” into an unique “Results and Discussion” section. Other changes to improve the readability of the paper are suggested below.
Title
It is too long.
Suggestion: “Cross-Recognition of Promoters by the Nine SigB Homologues present in Streptomyces coelicolor A3(2)”
Introduction
- The authors did not mentioned at all the phenomenon of “Programmed Cell death” that is well known and important in Streptomyces in their introduction?
- The sentence “In fact, transcription of six genes encoding these SigB homologues (SigB, SigH, SigI, SigK, SigL, SigM) was induced by osmotic stress in S. coelicolor A3(2) [49].” that appears late in the Discussion should perhaps appear earlier in the introduction and discussed.
Results
I would strongly recommend to the authors to start their paper by the section entitled
2.1. Levels of SigB and SigH after osmotic stress and during differentiation.
Do the polyclonal antibodies obtained for SigB and SigH cross react?
The authors mention the proteolytic degradation of UshX in liquid culture but was the transcriptional situation of this region established in this condition? Is UshX really transcribed in such condition?
In Figure 5b (now 4b), the situation is rather clear for SigH and UshX but is far less clear for SigB. To what correspond the two major bands seen with the antibody against SigB? Please indicate by an arrow the position of SigB in Figure 5b (now 4b), middle picture.
“2.2. Deletion of the sigB and sigH operons in S. coelicolor A3(2)”
This section should come in second and should be considerably shortened. Figure 1 and 3 should go into supplementary data as well as perhaps the poorly quantitative and thus poorly informative figures 2 and 4. The discrepancy between the results of the authors and those previously published by the Roe’s group more likely rely on the different genetic background than on the strategy used to disrupt the genes. The lengthy discussion of the differences between their paper and previous published work by Roe’s group weakens the paper of the authors and should be shortened.
The concomitant deletion of several sigma factors involved in the osmotic stress response would probably necessary to have a noticeable effect of osmotic stress on S. coelicolor biomass yield.
2.3. Identification of SigB regulon using a heterologous two-plasmid system
2.4. Cross-recognition of promoters by nine homologues of SigB
- This part is interesting but as no quantitative data are provided, one cannot appreciate the difference of affinity of the different sigma factors for the different promoters.
- Figure 9b: the blue of this table is far too strong. It impedes reading and does not need to be colored.Please distinguish with a color code what are the SigB, SigF, SigG and SigH dependent promoters.
Discussion
The discussion is far too long and is a repetition of many parts of the results section. It is thus recommended to condense the sections “Results” and “Discussion” into an unique “Results and Discussion” section. This study demonstrated the cross recognition of the different promoters by more than one and up to five sigma factors but in condition of overexpression of the sigma factors in the heterologous host, E. coli. The latter’s are thus not interacting with their usual core RNA polymerase. The authors could not establish a clear consensus promoter sequence specific for each sigma factor but this might be due to the limited number of promoter fragments under study. However such specificity, promoting at least different affinity of various sigma factor for a promoter region, ought to exist at least for SigI that recognizes a single promoter (SCO2026). The length and sequence of the spacer between -10 and -35 regions may play a role in this specificity/affinity in the relation with the distance between the 2.4 and 4.2 region of the sigma factors?
As partially done for SigB and SigH (Figures 5 and 8), it would be interesting to follow the expression of these promoters in the over-expressing and disruptive (single or double) mutants of the genes encoding the various SigB-like of S. coelicolor, throughout growth and in condition of low or high osmolarity. This would provide information on the temporality of expression of the different sigma factors, on the nature of their targets and on their involvement in the resistance to osmotic stress and/or differentiation. This may help to answer to the question of the biological significance of the high redundancy of these sigma factors recognizing similar promotor sequences. A ChIP on Chip approach may also be fruitful to have a more complete view of the situation? At last, it would be nice if the authors can speculate a little bit on the nature of the link existing between osmotic stress and morphological differentiation? Is it linked to the degradation of glycogen that increases the turgor pressure necessary for the raising of aerial mycelium? Is it linked to the accumulation of metabolites, not used for anabolism, because of growth slow down ?
Editorial comments
Line 69 “…via the RsbT activator…
Line 160 “ The transcriptional activity of the identified promoted was tested in vivo …”
Line 459 “…compared to all other sigma factors…”
Line 494 “ These two sigma factors…”
Line 564 “We attempted…”
Line 566 “…may contribute to the elucidation of their role…”
Line 755 “…where the levels of both UshX and short versions of SigH gradually increase towards later stages of development.”

Reviewer 3 Report
The manuscript of Sevcikova et al. focuses in the identification of different promoters recognized by sigB from Streptomyces coelicolor and in the description of a cross-recognition with other S. coelicolor sigB homologues. The expression pattern of SigB and SigH, a sigB homologue, is also studied by immunodetection with specific antibodies.
The manuscript presents a wide work and complements previous publications from this group.
The manuscript needs some corrections to be acceptable for publication. For example, some parts of the introduction are written in very long paragraphs that complicate the reading and understanding. A similar comment applies to the discussion, which also has very long paragraphs and some parts are repeated from the results. So, from my point of view, rewriting this part may give more strength to the manuscript.
My comments
Considering that NepA has the unique promoter known, up to now, as recognized by SigN, have the authors considered to use the two plasmid system to identify new promoters recognized by this sigma unit?
Have the authors considered the possibility of getting mutants in two or more sigma factors? (For example the double mutant SigB-SigH).
Taking into account that any phenotype is observed in the mutant generated in sigB and SigH, figures 2 and 4 may be more suitable for the supplementary materials.
Please eliminate the first sentence of the introduction L33-34.
Figure 2: please change the red color of the letters that indicate the name of the different strains. Other possibility for figures 2 and 4 is to put an empty plate on the side with the names of the strains used.
Figure 3: the restriction site EcoRI is not drawn in the wild type strain, so the reader cannot identify the fragment originated by the digestion BamHI- EcoRI in the wt strain.
In the legend, please change S. coelicolor A3(2) by S. coelicolor M145.
L281-286, the authors state that the antisigma protein UshX is proteolytically degraded in these conditions. Do the authors have some evidence that this gene is expressed under these conditions? Example RT-PCR.
L283 the reference to Figure 2A is wrong, it must be Figure 5A. The same occurs in L327, where the reference to Figure 2B must be changed by Figure 5B.
Part 2.3 line 240
The authors make the screening of two different libraries to identify the SigB recognized promoters. Is any of the promoters isolated in both libraries?
Figure 5
For the reader it is not clear whether NMP is the same as MM and it appears that the only difference is that one is liquid, whereas the other is solid. Please clarify along the manuscript.
The antibodies obtained recognize very strongly unspecific bands. Have the authors identified some of these bands? In some Westerns the unspecific bands are much more intense that the bands obtained with the studied protein, for instance in the case of anti SigB antibodies. So, in the Western from solid media it is difficult to observe the SigB band.
The central panel of Figure 5B may be eliminated and it should be indicated in the text that no protein SigB was detected under normal solid grown. Please use the same order of figures in Figure 5A and 5B.
Do the authors have some explanation to the observation that much more SigB is produced under stress conditions in liquid media than in solid media? Due to this different result in liquid and solid media, the authors must clarify the stress conditions used to get the RNA used for the results presented in Figure 8 (lines 418-421).
If possible, please use the same order in the promoters for table 1, Figure 7B and Fig 8.
L 458 the word “this” is repeated.
L466 The authors make a reference to Figure 1A and must be 9A.
Fig 9. If possible improve the colors of Figure 9A. Change the color of the blue boxesin Figure 9B, it is very hard to see the name of the plasmids. The Figure 9B must be improved by indicating the original sigma factor that allowed for the identification of the different promoters.
According to the results of Figure 9B, the promoter SCO1566 is just recognized by SigB. However, in Figure 8 there is transcription in the mutant SigBop. Do the authors have some explanation for this result?
Round 2
Reviewer 2 Report
see the attached file.
